# The Structure and Crystallizing Process of NiAu Alloy: A Molecular Dynamics Simulation Method

Dung Nguyen Trong [1],*, Van Cao Long [1] and Ştefan Ţălu [2]

1   Institute of Physics, University of Zielona Góra, 65-516 Zielona Góra, Poland; caolongvanuz@gmail.com
2   The Directorate of Research, Development and Innovation Management, Technical University of Cluj-Napoca, 15 Constantin Daicoviciu St., Cluj-Napoca, 400020 Cluj County, Romania; stefan_ta@yahoo.com
*   Correspondence: dungntsphn@gmail.com

**Abstract:** This paper studies the influence of factors such as heating rate, atomic number, temperature, and annealing time on the structure and the crystallization process of NiAu alloy. Increasing the heating rate leads to the moving process from the crystalline state to the amorphous state; increasing the temperature (T) also leads to a changing process into the liquid state; when the atomic number (N), and t increase, it leads to an increased crystalline process. As a result, the dependence between size (l) and atomic number (N), the total energy of the system ($E_{tot}$) with N as $l \sim N^{-1/3}$, and $-E_{tot}$ always creates a linear function of N, glass temperature ($T_g$) of the NiAu alloy, which is $T_g = 600$ K. During the study, the number of the structural units was determined by the Common Neighborhood Analysis (CNA) method, radial distribution function (RDF), size (l), and $E_{tot}$. The result shows that the influencing factors to the structure of NiAu alloy are considerable.

**Keywords:** annealing time; crystallize process; molecular dynamics; NiAu alloy; structure

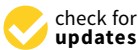

## 1. Introduction

Today, the alloys PtAu [1], PdAu [2], NiAu [3] are receiving great attention from theoretical and experimental scientists [4,5] because they have many special properties compared to pure materials [6,7]. In particular, NiAu alloy is synthesized by two metals, Ni and Au, and applied in many fields of science, technology, and life such as magnetism [8–10], photocatalyst [11,12], DNA markers [13,14], or cancer treatment [15] as the agent in cell separation [16,17], and biological processing [18,19] which increase contrast and biological agents [20]. The properties of alloys such as ionization, optics, and magnetism [21] depend on the shape and the concentration of doping [22].

There are a lot of methods to research and manufacture NiAu alloy, such as experimental, theoretical, and simulated. The experimental method includes mechanical grinding [23], electric arc [24], deposition [25,26], electrochemical [27], hydrothermal [28], Sol-Gel [29], mechanics [30], micro-emulsion [31], and colloidal solution [32].

These methods can change the size and shape of the alloy in normal conditions and do not require an environment of pressure (P) and high temperature (T) [33]. Theoretical methods include initial principles, Ab initio model [34], and methods of Molecular Dynamics (MD) simulation [35–37] combined with different interaction potentials such as Finnis–Sinclair (FS) [38], and Sutton–Chen (SC) [39,40]. In particular, the method of MD simulation is considered as the most preeminent method today with low research costs, capable of researching at the atomic level and providing a huge amount of information o the structure and explaining relevant physical mechanisms [41,42].

The result of research of Ni, Au metal and NiAu alloy in the liquid state, crystalline state, an amorphous state [43–45] shows that at the temperature (T), T = 300 K, pure Ni, Au metals do not change the structure transition process when being combined to form NiAl alloy; their electronic mobility is in ranges from 5% to 99% depending on the impurity concentration [46] that leads to the crystallize processes, and the structural

transitions occurring quickly [47]. They use NiAu alloys as catalysts for clean water [48–50] by using Au atoms in combination with Ni atoms to ionize water atoms. However, to meet below phase diagram, the study and synthesis of NiAu alloy [51] are being performed by electrolysis, the results showing an irregularly distributed shape and a medium particle size, which is 25 nm [52].

Recently, researchers have proposed a method using low temperature [53,54] and reduction as a way to synthesize NiAu alloy [55]. Vasquez et al. [56] also used this method to synthesize $Au_3Fe$, $Au_3Co$, and $Au_3Ni$ alloys. With this one, the shape and size of NiAu alloy are better controlled, and it has carried out more research in recent years [57–61]. With these obtained results [3], at high pressure [62,63] and Morse potential interaction, we can measure accurately the elastic modulus of AuNi. Lecadre et al. [64] studied the scattering and diffusion mechanism in Au-Ni alloys with $Au_3Ni$ and $Au_3Ni_2$ ratios. Berendsen et al. [65] have identified the transition temperature ($T_m$) of NiAu ranges from $T_m$ = 1100 K to $T_m$ = 1300 K with Au impurity concentration of 58% [66]. Combined with our recent results as Al [67], FeNi [68], AlNi [69], $Ni_{1-x}Fe_x$ [70], $Ni_{1-x}Cu_x$ [71], Ni [72,73], these results show that the transition temperature ($T_m$) of Ni material; $T_m$ is always proportional with atom number (N), $N^{-1/3}$ [74,75], and the electronic structure of AuCu [76] and AgAu [77]. The phase transition of Ni material can be determined by stress or temperature [78–81], and the bonding length of Ni-Ni determined by the experimental method is r = 2.43 Å [82], while the simulation method of Dung, N.T is r = 2.45 Å [73], and P.H. Kien is r = 2.52 Å [72]. Meanwhile, Ni and Au both have significant differences in atomic radius (R) sizes such as: Ni is R = 1.245 Å, Au is R = 1.44 Å, and surface energy (E) of Ni is E = 149 $meVÅ^{-2}$, Au is E = 96.8 $meVÅ^{-2}$ [83], which lead to the diffusion of Au atoms in the crust and Ni atoms in the core layer [84]. So, what processes were happened to NiAu alloy when there was a change in heating rate, atomic number, and temperature? To answer this question, we focus on studying the factors that affect the structure and crystallization process of NiAu alloys.

## 2. Method of Calculation

Initially, the ratio between NiAu alloy and Ni:Au is 1:1, as in 2048 NiAu atoms, there are 1024 Ni atoms, 1024 Au atoms ($NiAu_{2048}$), 2916 atoms ($NiAu_{2916}$), 4000 atoms ($NiAu_{4000}$), 5324 atoms ($NiAu_{5324}$), 6912 atoms ($NiAu_{6912}$); all samples are studied by molecular dynamics (MD) simulation method [85–95] with embedded Sutton–Chen (SC) interaction [39,96–99] and boundary conditions recirculating with the Equation (1):

$$E_{tot} = \sum_{i=1}^{N} \frac{1}{2} \sum_{j=1,j\neq i}^{N} \Phi(r_{ij}) + F(\rho_i), \; \Phi(r_{ij}) = \varepsilon \left(\frac{a}{r_{ij}}\right)^n, \; F(\rho_i) = -\varepsilon C \sum_{i=1}^{N} \sqrt{\rho_i}, \; \rho_i = \sum_{j=1,j\neq i}^{N} \rho(r_{ij}), \; \rho(r_{ij}) = \left(\frac{a}{r_{ij}}\right)^n \quad (1)$$

The parameters of the NiAu alloy (Table 1) are shown below.

**Table 1.** Parameters of NiAu alloy.

| Alloy | $\varepsilon$ ($\times 10^{-2}$ eV) | a (Å) | n | m | C |
|-------|------|------|-----|-----|--------|
| Ni | 1.5707 | 3.52 | 9 | 6 | 39.432 |
| Au | 1.2793 | 4.08 | 10 | 8 | 34.408 |
| NiAu | 1.4175 | 3.80 | 9.5 | 7.0 | 36.834 |

The parameters of the alloy are determined by the mathematical Formula (2):

$$\varepsilon_{NiAu} = \sqrt{\varepsilon_{Ni} \cdot \varepsilon_{Au}}; \; a_{NiAu} = \frac{(a_{Ni} + a_{Au})}{2}; \; n_{NiAu} = \frac{(n_{Ni} + n_{Au})}{2}; \; m_{NiAu} = \frac{(m_{Ni} + m_{Au})}{2}; \; C_{NiAu} = \sqrt{C_{Ni} \cdot C_{Au}} \quad (2)$$

At all samples, there is an increase in temperature (T) from T = 0.0 K to T = 2000 K to NiAu alloy at the liquid state. From the liquid state, the temperature of the samples was reduced from T = 2000 K to T = 300 K to change from a liquid state to a crystalline one. After getting NiAu alloy, $NiAu_{6912}$ alloys are run MD with a heating speed of $4 \times 10^{11}$ K/s, $4 \times 10^{12}$ K/s, $4 \times 10^{13}$ K/s, $4 \times 10^{14}$ K/s at (T), T = 300 K. After determining the heating speed of $4 \times 10^{12}$ K/s to be appropriate, the effects of $NiAu_{2048}$, $NiAu_{2916}$, $NiAu_{4000}$, $NiAu_{5324}$, $NiAu_{6912}$ at T = 300 K; $NiAu_{6912}$ at T = 300 K, 400 K, 500 K, 600 K, 700 K, 900 K, 1100 K are studied. All given samples are structurally studied through shape, size (l) as (3),

$$\rho = \frac{N}{V} \rightarrow 1 = \sqrt[3]{\frac{N}{\rho}} = \sqrt[3]{\frac{(m_{Ni} \cdot n_{Ni} + m_{Au} \cdot n_{Au})}{\rho}} \tag{3}$$

radial distribution function (RDF) as (4):

$$g(r) = \frac{V}{N^2} \left\langle \frac{\sum_i n_i(r)}{4\pi r^2 \Delta r} \right\rangle \tag{4}$$

In it: 1, ρ, r, N, $n_i(r)$, V, g(r) is the size, density, radial distance, the number of atoms, the coordinates, the volume, the probability of finding an atom in the distance from r to r + Δr. To determine the number of structural units, are applied the Common Neighborhood Analysis (CNA) method [100–103]. The crystallizing process is carried out based on the laws of Nosé el [104] and Hoover el [105] and uses the techniques of particle size analysis, atomic composition, and configuration [106].

### 3. Results and Discussion

#### 3.1. Effect of Heating Rate

The factors that affect the heating rate $4 \times 10^{12}$ K/s, $2 \times 10^{13}$ K/s, $4 \times 10^{13}$ K/s, $2 \times 10^{14}$ K/s, and $4 \times 10^{14}$ K/s on the structural characteristics and crystallization process of $NiAu_{5324}$ alloy at temperature (T), T = 300 K, are shown in Figure 1.

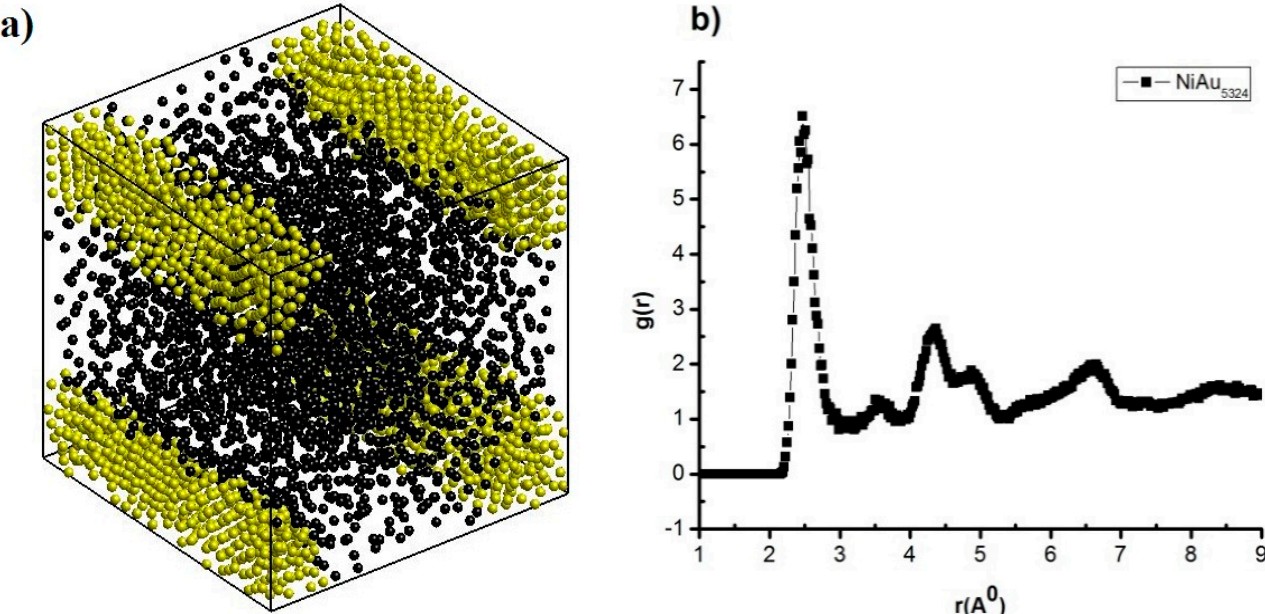

**Figure 1.** The shape (**a**), g(r) of $NiAu_{5324}$ alloy (**b**) at T = 300 K, and heating rate $4 \times 10^{12}$ K/s.

The result shows that when NiAu$_{5324}$ alloy at T = 300 K with the heating rate of $4 \times 10^{12}$ K/s, it has a cube shape, made by two atoms: Ni shown in black and Au in yellow (Figure 1a), and has structural features such as r of radial distribution function (RDF) = 2.47 Å; the height of RDF is g(r) = 6.51, size (l), l = 10.16 nm, E$_{tot}$ = −446.09 eV (Figure 1b). That increasing heating rate from $4 \times 10^{12}$ K/s to $2 \times 10^{13}$ K/s, $4 \times 10^{13}$ K/s, $2 \times 10^{14}$ K/s, and $4 \times 10^{14}$ K/s leads to r decreases from r = 2.47 Å to r = 2.43 Å and g(r) decreases from g(r) = 6.51 to g(r) = 6.05, l negligibly changes from l = 9.71 nm to l = 10.76 nm, and E$_{tot}$ negligibly changes from E$_{tot}$ = −440.25 eV to E$_{tot}$ = −447.18 eV (Table 2). These results show that increasing the heating rate leads to NiAu$_{5324}$ alloy change, the state from crystalline to amorphous. To study the process of structural transition, the CNA method was used and the results are shown in Figure 2.

**Table 2.** The structural features such as r, g(r) of the radial distribution function, l, and E$_{tot}$ with t different heating rates.

| Heating Rates (K/s) | $4 \times 10^{12}$ | $2 \times 10^{13}$ | $4 \times 10^{13}$ | $2 \times 10^{14}$ | $4 \times 10^{14}$ |
|---|---|---|---|---|---|
| r (Å) | 2.47 | 2.47 | 2.45 | 2.45 | 2.43 |
| g (r) | 6.51 | 6.08 | 5.43 | 4.87 | 6.05 |
| l (nm) | 10.16 | 10.35 | 10.05 | 9.71 | 10.76 |
| E$_{tot}$ (eV) | −446.09 | −446.84 | −447.18 | −447.15 | −440.25 |

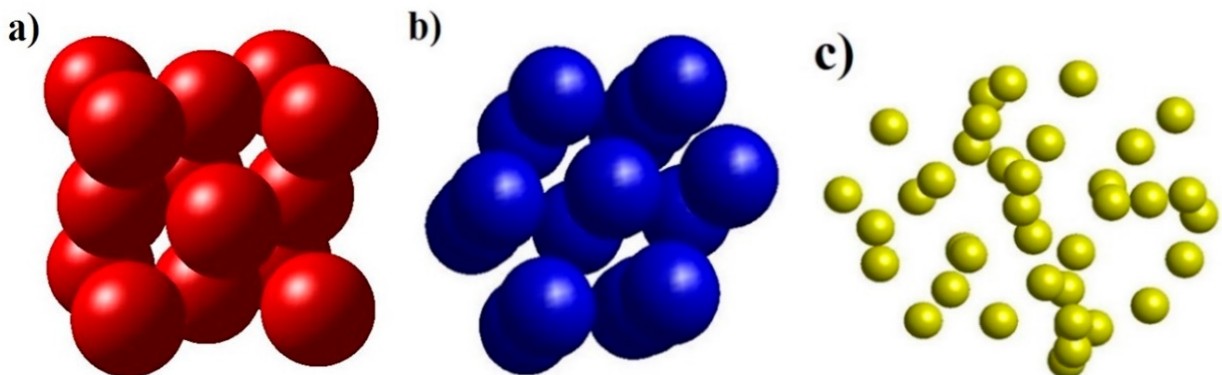

**Figure 2.** The structural unit number shape includes FCC structure (**a**), HCP structure (**b**), Amor structure (**c**) of NiAu alloy.

The result shows that NiAu$_{5324}$ alloy at the heating rate of $4 \times 10^{12}$ K/s has structural shapes (Figure 3a) corresponding with 03 links Ni-Ni, Au-Au: Ni- Ni is r = 2.47 Å, Ni-Au is r = 2.47 Å, Au-Au is r = 3.17 Å (Figure 3b), and expresses through structural unit number FCC (Figure 2a), HCP (Figure 2b), Amor (Figure 2c). The obtained results are consistent with the results of Ni-Ni by experimental methods r = 2.43 Å [82], and R = 1.245 Å, with the simulation method r = 2.45 Å [73], r = 2.52 Å [72], for Au-Au, only X-ray diffraction results in an atomic radius value R = 1.44 Å [83,84]. Increasing heating rate from $4 \times 10^{12}$ K/s to $2 \times 10^{13}$ K/s, $4 \times 10^{13}$ K/s, $2 \times 10^{14}$ K/s, and $4 \times 10^{14}$ K/s leads to r of link Ni-Ni, Ni-Au, Au-Au change values. Besides, when Ni-Ni changes from r = 2.47 Å to r = 2.41 Å, Ni-Au decreases from r = 2.47 Å to r = 2.43 Å, Au-Au decreases from r = 3.17 Å to r = 3.09 Å, corresponding to the change of g(r) and structural unit number FCC, HCP, Amor; as FCC decreases from 802 to 0.0 FCC, HCP decreases from 811 HCP to 13 HCP, Amor increases from 3711 Amor to 5311 Amor (Table 3). That confirms that there is an increase in the heating rate when the crystallization process decreases.

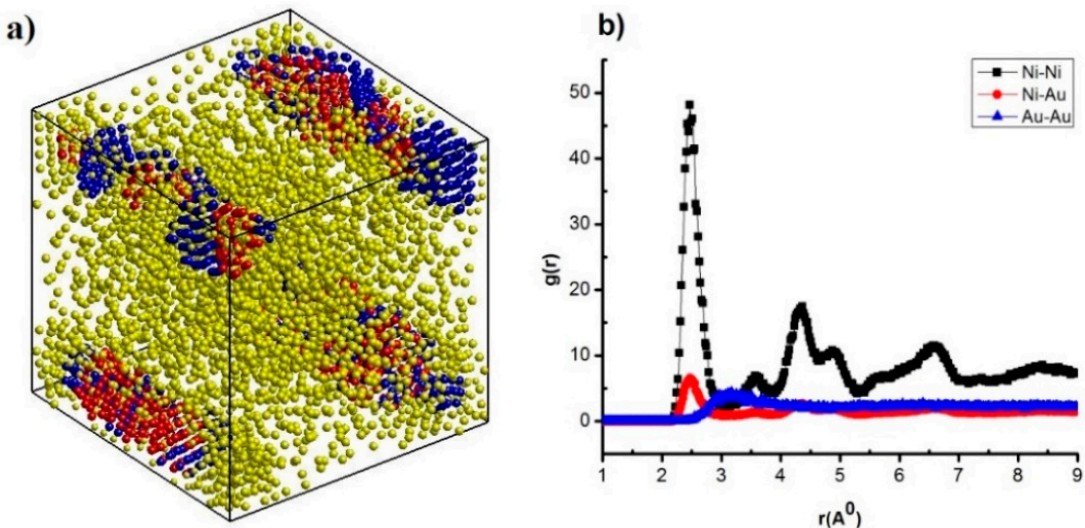

**Figure 3.** The structural shape (**a**), radial distribution function (**b**) of NiAu$_{5324}$ alloy at heating rates of $4 \times 10^{12}$ K/s.

**Table 3.** The structural features such as links Ni-Ni, Ni-Au, Au-Au include r and g(r) with a different atomic number (N).

| NiAu$_{5324}$ Alloy | r(A$^0$) | | | g(r) | | | Structural Unit Number | | |
|---|---|---|---|---|---|---|---|---|---|
| | $r_{Ni-Ni}$ | $r_{Ni-Au}$ | $r_{Au-Au}$ | $g_{Ni-Ni}$ | $g_{Ni-Au}$ | $g_{Au-Au}$ | FCC | HCP | Amor |
| $4 \times 10^{12}$ | 2.47 | 2.47 | 3.17 | 48.11 | 6.51 | 4.41 | 802 | 811 | 3711 |
| $2 \times 10^{13}$ | 2.45 | 2.47 | 3.11 | 44.98 | 6.08 | 4.35 | 111 | 139 | 5074 |
| $4 \times 10^{13}$ | 2.41 | 2.45 | 3.11 | 39.84 | 5.43 | 4.75 | 58 | 115 | 5151 |
| $2 \times 10^{14}$ | 2.45 | 2.45 | 3.11 | 33.60 | 4.87 | 5.17 | 0 | 26 | 5298 |
| $4 \times 10^{14}$ | 2.45 | 2.43 | 3.09 | 42.15 | 6.05 | 5.70 | 0 | 13 | 5311 |
| Results simulation experiment | 2.45 [73] 2.52 [72] 2.43 [82] | | 2.88 [83,84] | | | | | | |

## 3.2. Effect of Atomic Number

Similarly, with the influence of atomic numbers NiAu$_{2048}$, NiAu$_{2916}$, NiAu$_{4000}$, NiAu$_{5324}$, and NiAu$_{6912}$ on structural characteristics, results are shown in Figure 4.

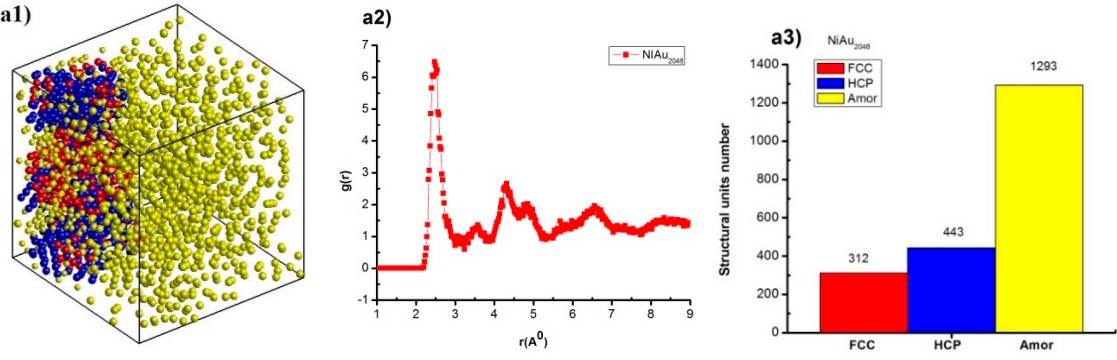

**Figure 4.** *Cont.*

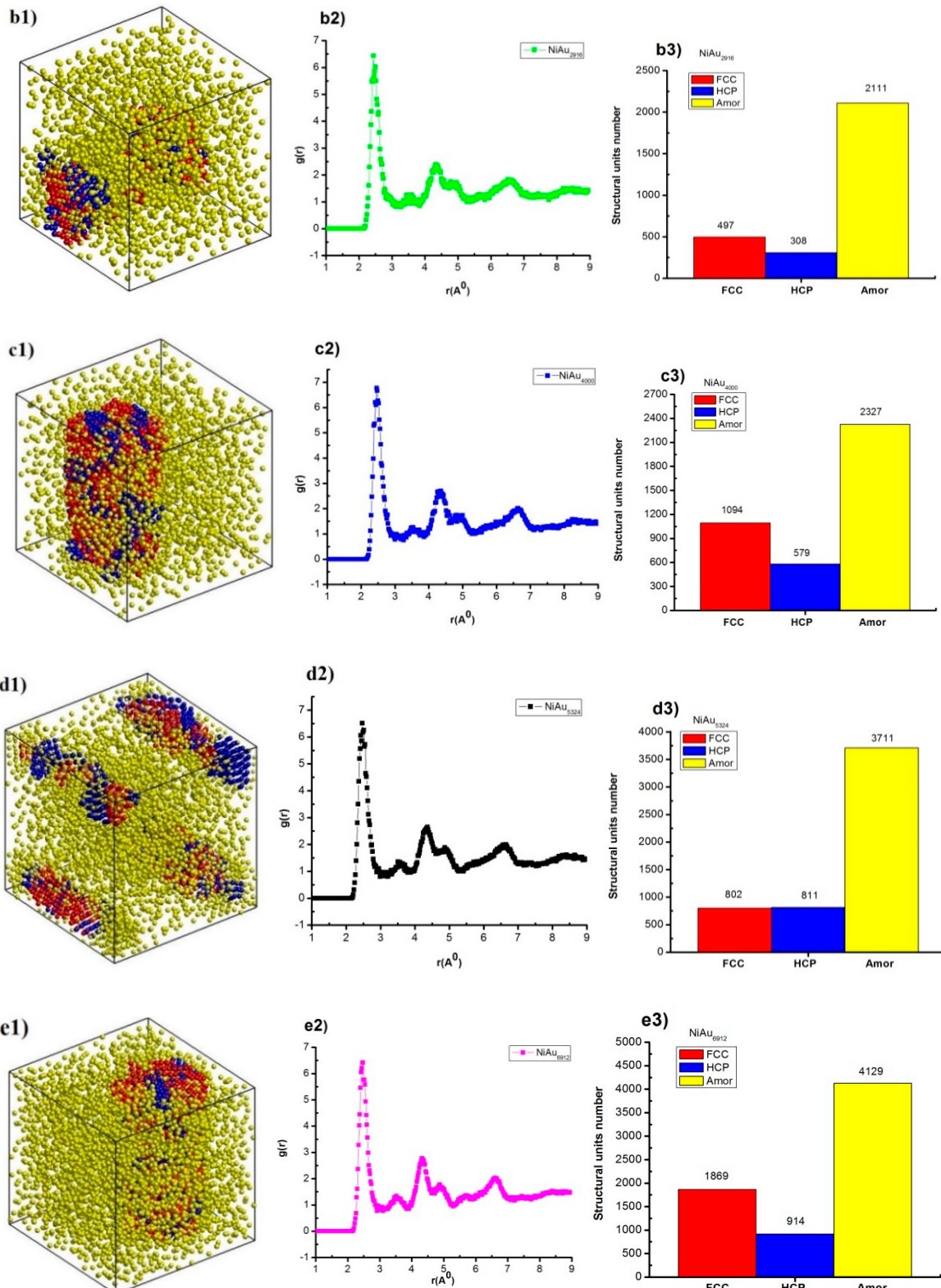

**Figure 4.** The structural shape (**a1,b1,c1,d1,e1**), radial distribution function (RDF) (**a2,b2,c2,d2,e2**), structural unit number (**a3,b3,c3,d3,e3**) of NiAu alloy with different N.

The results indicate that $NiAu_{2048}$ alloy at T = 300 K has structural shape and l = 7.34 nm, $E_{tot}$ = −173.85 eV (Figure 4(a1)), together with the radial distribution function, has r = 2.49 Å, g(r) = 6.47 (Figure 4(a2)), and the structural unit numbers are 312 FCC, 443 HCP, 1293 Amor (Figure 4(a3)). When increasing atoms number from $NiAu_{2048}$ to $NiAu_{2916}$, $NiAu_{4000}$, $NiAu_{5324}$, $NiAu_{6912}$, then structural shape changes, size (l) increases from l = 7.34 nm to l = 10.89 nm, $E_{tot}$ decreases from $E_{tot}$ = −173.85 eV to $E_{tot}$ = −580.35 eV (Figure 4(b1,c1,d1,e1)); r changes from r = 2.49 Å to r = 2.45 Å, g(r) changes from g(r) = 6.43 to g(r) = 6.77 (Figure 4(b2,c2,d2,e2)); and the structural unit number change corresponding to FCC about from 312 FCC to 1869 FCC, HCP about from 308 HCP to 914 HCP, Amor increases from 1293 Amor to 4129 Amor (Figure 4(b3,c3,d3,e3)). The given result indicates that increasing N leads to l increase and $E_{tot}$ decrease. As a result, there is a relationship between l, N, and $E_{tot}$, N. To confirm the defining relationships between size, atom number, and between $E_{tot}$, atom number of NiAu alloy, the results are shown in Figure 5.

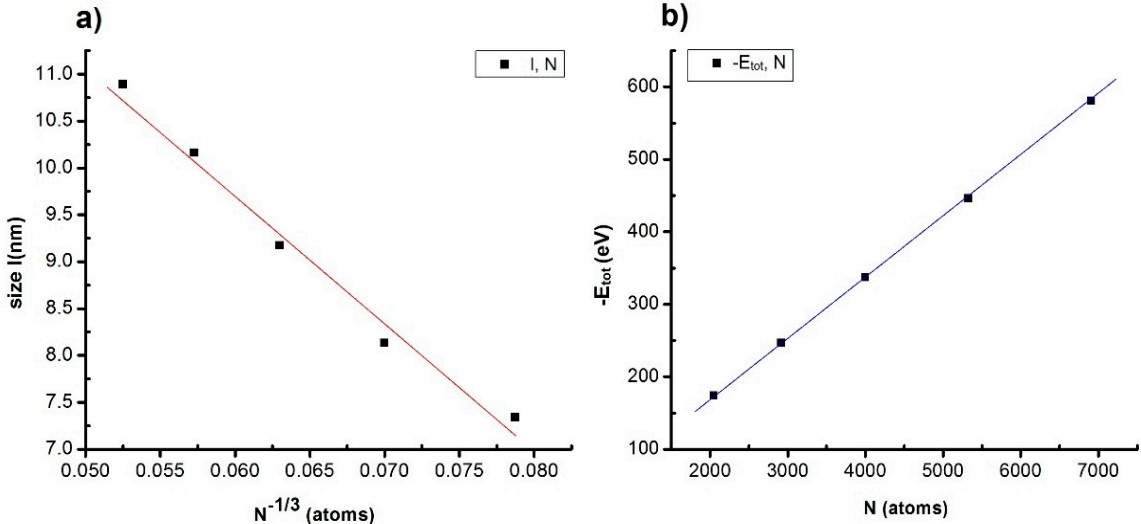

**Figure 5.** The relationship between characteristic quantities such as structures as dependence between l, N (**a**); −$E_{tot}$, N (**b**) of the NiAu alloy with different N.

The given result shows that increasing N leads to l increase, and satisfy the formula: l = 18.021 − 138.153$N^{-1/3}$, corresponding to l~$N^{-1/3}$ (Figure 5a) and −$E_{tot}$ proportional to N (Figure 5b). The results are consistent with the results of crystallizing process temperature ($T_m$) proportional with $N^{-1/3}$ [74,75] and size (l or D) proportional with $N^{-1/3}$ [67–73]. This proves that increasing atom number leads to crystalline atoms number FCC, HCP increase, Amor decrease, size increase, the total energy of system decrease; and the relationship with l~$N^{-1/3}$ is an important result for future experimental implementation.

### 3.3. Influence of Temperature

The research results of the effect of temperature, T = 300 K, 400 K, 500 K, 600 K, 700 K, 900 K, 1100 K 1300 K, and 1500 K, on the structural characteristics are shown in Figure 6.

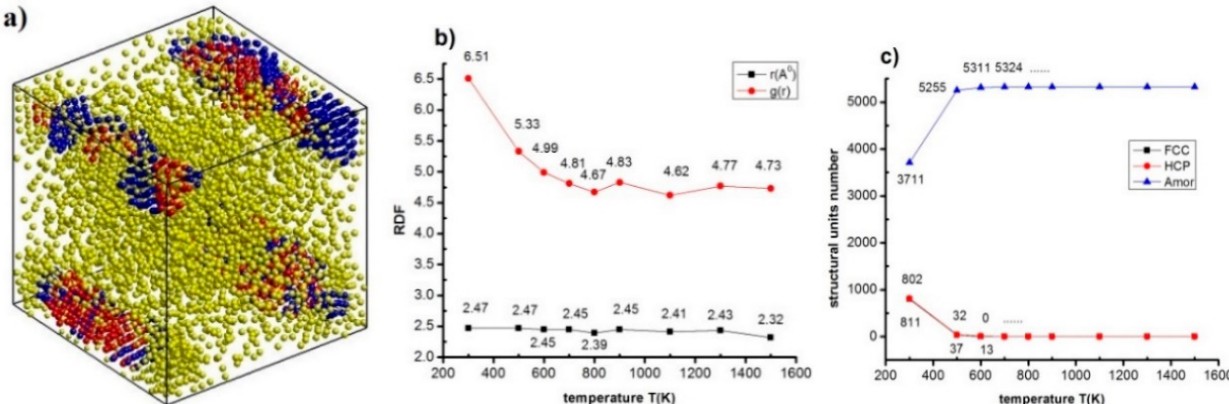

**Figure 6.** The structural characteristics as structural shape (**a**) of NiAu$_{5324}$ alloy at T = 300 K; RDF (**b**) and structural unit number (**c**) at different T.

The results indicate at T = 300 K, NiAu$_{5324}$ alloy has structural shape, r = 2.47 Å, g(r) = 6.51, 802 FCC, 811 HCP, 3711 Amor (Figure 6a). When increasing T from T = 300 K to T = 1500 K, r decreases from r = 2.47Å to r = 2.32 Å, g(r) changes from g(r) = 6.51 to g(r) = 4.62 (Figure 6b), FCC decreases rapidly from 802 FCC to 0.0 FCC, HCP decreases dramatically from 811 HCP to 0.0 HCP, Amor increases from 3711 HCP to 5324 HCP, at T = 700 K Amor state maximum increases (Figure 6c). This proves that with T < 700 K, NiAu alloy is in the crystalline state, T > 700 K, and NiAu alloy is in the liquid state, thereby, this is a crystallizing process between the crystallization and liquid states of NiAu alloy. The results of phase transition from crystalline state to a liquid state by type 1 phase transition theory, for each value of temperature (T), will correspond to a total energy value of the system (E$_{tot}$). To confirm that, a study of the relationship between T and E$_{tot}$ was carried out and the obtained results are shown in Figure 7.

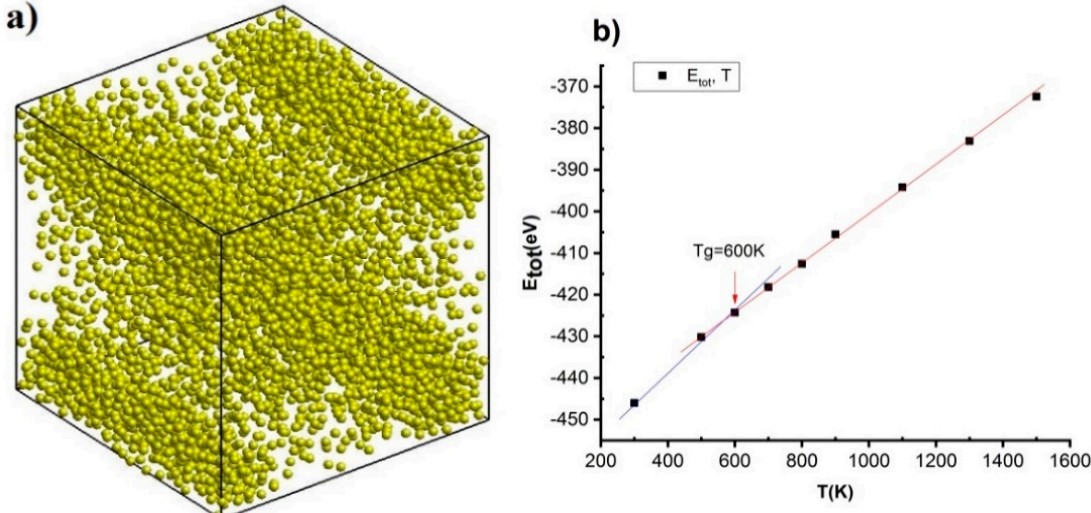

**Figure 7.** The structural shape of NiAu$_{5324}$ alloy at temperatures 700 K (**a**), phase transition (**b**) of NiAu$_{5324}$ alloy at different temperatures.

Increasing T from T = 300 K to T = 400 K, 500 K, 600 K, 700 K, 900 K, 1100 K, 1300 K, and 1500 K leads to l insignificant change range l = 7.34 nm to l = 7.35 nm and E$_{tot}$ increase from E$_{tot}$ = −446.0 eV to E$_{tot}$ = −430.2 eV, −424.3 eV, −418.2 eV, −415.6 eV, −405.5 eV, −394.2 eV, −383.1 eV, and −372.4 eV. In it, the structural shape of NiAu alloy at T = 700 K (Figure 7a) and an interrupting point at T = 600 K then corresponds with E$_{tot}$ = −418.2 eV observed at glass temperature (T$_g$), T$_g$ = 600 K (Figure 7b). The results obtained show that,

with T < 600 K, NiAu$_{5324}$ alloy exists in the crystalline state and T > 600 K is in a liquid state, this result is completely consistent with the results obtained in on and are considered as the basis for future empirical research.

### 3.4. Influence of Annealing Time

The crystallization process of NiAu$_{5324}$ alloy is carried out after the tempering period and shown in Figure 8, Table 4.

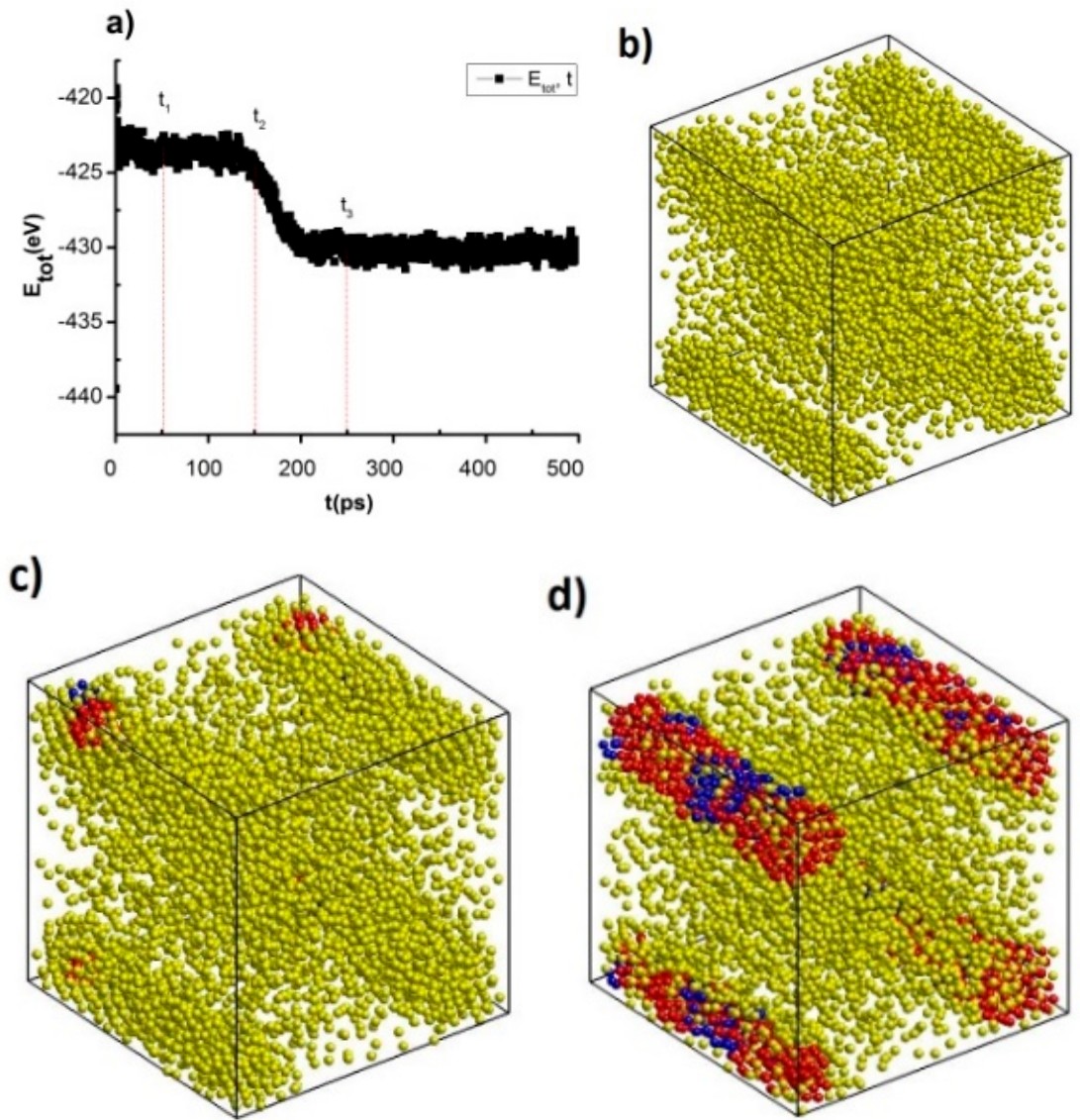

**Figure 8.** The crystallization process of NiAu$_{5324}$ alloy after different annealing time (**a**), the shape of NiAu alloy at t$_1$ = 50 ps (**b**), t$_2$ = 150 ps (**c**), t$_3$ = 250 ps (**d**).

**Table 4.** The structural characteristics and number of structural units of NiAu$_{5324}$ alloy after different annealing times.

| t(ps) | 50 | 150 | 250 |
|-------|------|------|------|
| r(Å) | 2.45 | 2.49 | 2.47 |
| g(r) | 5.26 | 5.08 | 5.88 |
| FCC | 0.0 | 58 | 1463 |
| HCP | 0.0 | 9 | 368 |
| Amor | 5324 | 5257 | 3493 |

The result shows that after incubation period, t = 50 ps, 150 ps, 250 ps, $E_{tot}$ decreases from $E_{tot}$ = −425 eV to $E_{tot}$ = −430 eV (Figure 8a), the structural shape at $t_1$ = 50 ps (Figure 8b), $t_2$ = 150 ps (Figure 8c), $t_3$ = 250 ps (Figure 8d) corresponding with the link length, the height of the RDF Ni-Au varies greatly from r = 2.45 Å to r = 2.49 Å, g(r) = 5.26 to g(r) = 5.88 and structural unit number FCC increases from 0.0 FCC to 1463 FCC, HCP increases from 0.0 HCP to 368 HCP, and Amor decreases from 5324 Amor to 3493 Amor (Table 4). The results obtained show that NiAu5324 alloy at Tg = 600 K after the annealing time t = 250 ps, the crystallization process increases, proving that NiAu5324 alloy is easy to crystallize at the temperature Tg and results obtained are consistent with the results of the previous alloys such as AlNi [69], NiCu [71], FeNi [68,70].

When increasing heating rate, the temperature leads to r, g(r) decreases, l, $E_{tot}$ increase, FCC, HCP decrease and Amor increase and vice versa with increasing atomic number and the incubation time at $T_g$ = 600 K. This is a very useful result for the process of implementing experimental results in the future.

## 4. Conclusions

After studying the effect of factors on the structure and crystallize process, we got the following results: The relationship between size (l) and atomic number (N) is determined according to the formula is $l \sim N^{-1/3}$; the total energy of the system ($-E_{tot}$) always depends on N by function, the result is consistent with results [67–75] when the glass temperature ($T_g$), $T_g$ = 600 K. With NiAu$_{5324}$, when increasing the heating rate from $4 \times 10^{12}$ K/s to $2 \times 10^{13}$ K/s, $4 \times 10^{13}$ K/s, $2 \times 10^{14}$ K/s, and $4 \times 10^{14}$ K/s leads to r decrease from r = 2.47 Å to r = 2.43 Å, and g(r) decrease from g(r) = 6.51 to g(r) = 6.05, corresponding to the change of g(r) and structural unit number. FCC, HCP, Amor FCC decreases from 802 FCC to 0.0 FCC, HCP decreases from 811 HCP to 13 HCP, Amor increases from 3711 Amor to 5311 Amor. When increasing atom number from NiAu$_{2048}$ to NiAu$_{2916}$, NiAu$_{4000}$, NiAu$_{5324}$, NiAu$_{6912}$, first peak position (r) of RDF changes values from r = 2.49 Å to r = 2.45 Å, g(r) changes from g(r) = 6.43 to g(r) = 6.77, the structural unit number changes corresponding to FCC increases from 312 FCC to 1869 FCC, HCP increases from 308 HCP to 914 HCP, Amor increases from 1293 Amor to 4129 Amor. Similarly, with NiAu$_{5324}$ when increasing T from T = 300 K to T = 1500 K, r decreases from r = 2.47 Å to r = 2.32 Å, g(r) changes in the range from g(r) = 6.51 to g(r) = 4.62, FCC decreases rapidly from 802 FCC to 0.0 FCC, HCP decreases rapidly from 811 HCP to 0.0 HCP, Amor increases from 3711 HCP to 5324 HCP, Amor state maximum increases. When increasing in annealing times at $T_g$ = 600 K leads to the structural unit number FCC, HCP increases, Amor decreases. It indicates that the heating rate increase leads to the NiAu alloy change from crystallizing state to the amorphous state; increasing atomic number, annealing times leads to crystallization process increase; increasing temperature leads to process change from a crystallization state to a liquid state. This is a very essential factor and a basis for future empirical research.

**Author Contributions:** D.N.T.: conceptualization, methodology, validation, investigation, writing—original draft preparation. V.C.L.: data curation. Ş.Ţ.: writing—review and editing. All authors have read and agreed to the published version of the manuscript.

**Funding:** This research received no external funding.

**Data Availability Statement:** The data that support the findings of this study are available from the corresponding author upon reasonable request.

**Conflicts of Interest:** The authors declare no conflict of interest.

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
