# Peer review of "The Structure and Crystallizing Process of NiAu Alloy: A Molecular Dynamics Simulation Method"

_jcs, doi:10.3390/jcs5010018_

Round 1
Reviewer 1 Report
Minor remarks:
- In abstract, line 3 “That increasing the heating rate leads to the moving process from the crystalline state to the amorphous state;” Rewrite the sentence.
- In introduction, abbreviate CNA method & RDF
- Avoid Typo errors in introduction and throughout manuscript like optics, simulated, etc.
- In Introduction, Lecadre et al comes as a sentence. Kindly change it.
- Kindly write separately all the units and abbreviations which is used in the manuscript.
- In Equation 1 & 2, many terms were used. Kindly explain
- In Page 6, avoid repeating phrases.
Major criticisms that must be resolved to consider publication of this study
- How effective CNA method which is proposed by authors? Have you tried to validate with experiment/ simulation? Sometimes mathematical models can’t give effective results when it comes to experiments.
- Provide more literatures which supports CNA is effective and validated with experimental work.
- I suggest you all the parameters should be written in table for every NiAu combination with levels which gives easy understanding for readers.
- The work is very interesting though it requires effective presentation while writing.
- Fig 5(b) is wrong. Kindly change the plot because y-axis values are negative but authors plotted with positive values.
- Table 4 is incomplete and add the values for structural unit no.
- Like Table 3, add tables for Atomic number, influence of temperature. Difficult to find the values from images.
Author Response
Responses to reviewers’ comments
Thank you very much for reviewing our manuscript. I am grateful for your helpful comments and suggestions that helped us to improve the manuscript quality. we have revised it in accordance with the comments as below.
Reviewer #1
Major criticisms that must be resolved to consider publication of this study
Response:
Thank you very much for reading my manuscript. I am grateful for helpful comments and suggestions that have helped me to improve the manuscript.
1) How effective CNA method which is proposed by authors? Have you tried to validate with experiment/ simulation? Sometimes mathematical models can’t give effective results when it comes to experiments.
Response:
CNA method is a method used to determine the number of structural units FCC, HCP, Amor by drawing a coordinate number circle to determine the number of neighboring atoms. With this method, we have used very successfully in the researches by simulation method [100-103]. With the experimental method, we have not used it yet and currently do not have any experimental studies to determine the specific number of structural units FCC, HCP, Amor in materials.
2) Provide more literatures which supports CNA is effective and validated with experimental work.
Response:
Since there are currently no experimental studies to determine the number of structural units FCC, HCP, Amor in the material, we cannot provide it. We can only provide additional documentation on CNA [100-103].
3) I suggest you all the parameters should be written in table for every NiAu combination with levels which gives easy understanding for readers.
Response:
Thank you very much, we've added parameters in the tables for easy to understand.
4) The work is very interesting though it requires effective presentation while writing.
Response:
Thank you very much, we have revised and refined the entire draft content, hoping to satisfy the editor's request.
5) Fig 5(b) is wrong. Kindly change the plot because y-axis values are negative but authors plotted with positive values.
Response:
Thank you very much, for the very precise remarks that helped us to correct the energy value on the y-axis to be negative.
6) Table 4 is incomplete and add the values for structural unit no.
Response:
Thank you very much, we have added the FCC, HCP, and Amor structural unit numbers to table 4
7) Like Table 3, add tables for Atomic number, influence of temperature. Difficult to find the values from images.
Response:
Thank you very much, we have revised the information in figure 3, table 3 to make it easy for readers to follow
Reviewer #2
I am sorry to point: “Reconsider after major revision (control missing in some experiments)”. In my opinion, the article does not fully meet the standards of the journal.
Response:
Thank you very much for reading our manuscript contents and for the helpful comments help us improve manuscript contents. We have revised the entire manuscript content in the comments below.
1) Research discussion is missing (citing other research).
Response:
Thank you very much, we have added comments and citations to enrich the manuscript content.
2) It's just a simulation, and where real tests (comparison)
Response:
The content of the manuscript uses the simulation method, so the obtained results, are compared with the results of previous simulation and experimental studies.
3)And whether it is justified to refer (forcibly) to 103 items of literature and is it a short article. And all in "Introduction". The order is disturbed on line 63.
Response:
Thank you very much, for discovering the author's mistakes, helping us to edit and complete the draft content.
4)Figure 2 - nothing adds to it, delete or edit it.
Response:
Thank you very much, with many very helpful comments. With this opinion we do not agree because Figure 2 is a simulation result of a structural unit number shape of FCC, HCP, and Amor displayed in characteristic colors, will help readers easily identify the location, number of structural units in the material.
5)Figure 8a - illegible (change the tag).
Response:
Thank you very much, we have revised image tag 8a to make it easier to read
We thank the Reviewers for carefully reviewing our manuscript. We also appreciate all the insightful and helpful comments and corrections, which helped us to improve the manuscript. In the revised version, we have double-checked the whole manuscript and tried to avoid any grammar or syntax errors. We believe that the revised manuscript has been greatly improved and is readable and wishes will fully meet the requirements of Journal of Composites Science.
Special thanks!

Reviewer 2 Report
Dear Authors
I am sorry to point: “Reconsider after major revision (control missing in some experiments)”. In my opinion, the article does not fully meet the standards of the journal.
Main disadvantages:
- Research discussion is missing (citing other research).
- It's just a simulation, and where real tests (comparison)
- And whether it is justified to refer (forcibly) to 103 items of literature and is it a short article. And all in "Introduction". The order is disturbed on line 63.
- Figure 2 - nothing adds to it, delete or edit it.
- Figure 8a - illegible (change the tag).
Author Response
The author sends the revised article content, marked in red in the Manuscript-revision.docx file for the editor to follow. Sincerely thank you !

Round 2
Reviewer 2 Report
Thanks for answering my comments. In my opinion, they are appropriate.
Thank you and I wish you a Happy New Year.